# Recombinant BCG Expressing the Subunit 1 of Pertussis Toxin Induces Innate Immune Memory and Confers Protection against Non-Related Pathogens

**DOI:** 10.3390/vaccines10020234

**Published:** 2022-02-03

**Authors:** Alex I. Kanno, Diana Boraschi, Luciana C. C. Leite, Dunia Rodriguez

**Affiliations:** 1Laboratório de Desenvolvimento de Vacinas, Instituto Butantan, São Paulo 05503-900, SP, Brazil; alex.kanno@butantan.gov.br (A.I.K.); luciana.leite@butantan.gov.br (L.C.C.L.); 2Shenzhen Institute of Advanced Technology (SIAT), Chinese Academy of Sciences (CAS), Shenzhen 518055, China; diana.boraschi@itb.cnr.it

**Keywords:** BCG, recombinant BCG, rBCG-S1PT, innate immune memory, trained immunity

## Abstract

BCG has shown the ability to induce protection against unrelated pathogens, which likely depends on an immune mechanism known as innate immune memory or trained immunity. In this study, we evaluated the induction of innate memory by a recombinant BCG strain expressing the genetically detoxified S1 subunit of the pertussis toxin (rBCG-S1PT). In vitro pre-exposure of naïve murine macrophages to rBCG-S1PT increased their innate/inflammatory response (IL-6, TNF-α, and IL-10) to a subsequent challenge with unrelated pathogens, as compared to pre-exposure to wild-type BCG. Following LPS challenge, mice immunized with rBCG-S1PT produced higher levels of IFN-γ, while the release of other inflammatory cytokines was comparable to that measured after BCG immunization. SCID mice previously immunized with rBCG-S1PT and challenged with pathogenic *Candida albicans* displayed a similar survival curve as BCG-immunized mice but a lower CFU burden in the kidneys, suggesting an innate memory-dependent control of *C. albicans* infection. This study highlights the potential of recombinant BCG to increase innate immune memory and, ultimately, non-specific protection, more effectively than wild-type BCG. To our knowledge, this is the first report describing the potential of a recombinant BCG strain to strengthen innate immune memory responses.

## 1. Introduction

The BCG vaccine has been shown to be highly protective against the development of tuberculosis (TB) in children. This centenary vaccine is in the vaccination programs of most developing countries. Different studies have shown that it protects children against severe forms of the disease; however, protection wanes with time, and adults are less protected against pulmonary TB. Several vaccine candidates are currently under clinical trials, including “improved” BCG-based recombinant vaccines (rBCG) that rely on the expression of antigens or immunomodulatory molecules to improve BCG immunogenicity and, consequently, protection induced against *Mycobacterium tuberculosis* [1].

Recombinant BCG expressing the genetically detoxified S1 subunit of the pertussis toxin (rBCG-S1PT) was demonstrated to induce protection in mice against *Bordetella pertussis* challenge. Its pathogen-specific response was consistently demonstrated in adult and neonate mice [2,3,4]. The distinct immunostimulating capacity of rBCG-S1PT was evident in different experimental models. For instance, rBCG-S1PT showed an enhanced inflammatory/innate activation and a greater anti-tumor effect in a mouse model of bladder cancer—a known clinical application of wild-type BCG [5,6]. In an experimental model of asthma, previous immunization with rBCG-S1PT enhanced Th1 lung immunity and down-modulated the allergic response induced by ovalbumin [7]. Additionally, rBCG-S1PT has shown to induce an increased inflammatory response in human peripheral blood mononuclear cells [8].

The first rBCG was generated about 30 years ago (reviewed in [9]). Its use against heterologous pathogens involved the expression of pathogen-specific antigens. On the other hand, the non-specific innate immune response induced by BCG has demonstrated protection against unrelated pathogens in an immunological phenomenon known today as innate immune memory or trained immunity [10,11]. Through this mechanism, innate cells such as macrophages and monocytes show an increased innate/inflammatory response against unrelated stimuli after a previous exposure to BCG. This involves an epigenetic reprogramming in the promoters of several inflammatory genes and a metabolic shift towards the glycolytic pathway. Protection against a fair number of unrelated pathogens, such as *Candida albicans* [12], *Streptococcus pneumoniae* [13], and H1N1 influenza virus, has been demonstrated [14]. Although BCG can modulate the immune system towards an increased response against heterologous aggressions, caution should be taken while interpreting these data. BCG can indeed increase the responsiveness to viral infections, but this does not always translate into protection [15]. Additionally, the experimental conditions that characterize the induction of innate memory in vitro need standardization [16].

Recombinant BCG strains are expected to induce trained immunity [17] in a similar fashion as wild-type BCG. rBCG strains expressing additional microbial antigens can display improved or new features in their capacity to induce innate immune memory responses. In this work, we provide experimental evidence, for the first time, that a recombinant BCG strain can afford enhanced innate memory response and protection against unrelated pathogens.

## 2. Materials and Methods

### 2.1. Microorganisms and Culture Conditions

Wild-type *M. bovis* BCG Moreau and previously generated rBCG-S1PT [4] were grown in Middlebrook 7H9 (Difco, Detroit, MI, USA) supplemented with 0.5% glycerol, 0.05% Tween 80 (Sigma-Aldrich^®^, Merck KGaA, St. Louis, MO, USA), and 10% OADC (oleic acid–albumin–dextrose–catalase; BBL, Cockeysville, MD, USA). Cultures were maintained at 37 °C and 5% CO_2_ until reaching an OD of 0.6. Then, the cells were washed with 10% glycerol and stored at −80 °C until use. *C. albicans* (ATCC 90112) was cultured in YPD (10 g/L yeast extract, 20 g/L peptone, and 20 g/L dextrose). *Staphylococcus aureus* (ATCC 25923) was cultured in LB (5 g/L yeast extract, 10 g/L tryptone, and 10 g/L NaCl) until reaching OD 0.6. *C. albicans* and *S. aureus* were washed with phosphate-buffered saline (PBS, 137 mM NaCl, 2.7 mM KCl, 8 mM Na_2_HPO_4_, and 1.5 mM KH_2_PO_4_ pH 7.2) and resuspended in 10% glycerol. Aliquots were stored at −80 °C until use. Before use, each microorganism was serially diluted and plated onto the respective media plates to evaluate the number of colony-forming units (CFU).

### 2.2. Mice and Ethics Statement

Five-to-seven-week-old female C57BL/6 and NOD/SCID mice were provided by Biotério Central do Instituto Butantan and Biotério Central da Faculdade de Medicina da Universidade de São Paulo (FMUSP), respectively. All procedures were performed according to the Instituto Butantan’s Ethics Committee of Animal Use (CEUA) and approved under protocol 5165040219.

### 2.3. Bone Marrow Extraction and Macrophage Differentiation

C57BL/6 mice were euthanized by CO_2_ using an appropriate chamber, and the tibia and femur were removed. Bone marrow cells were flushed with RPMI-1640 (Gibco, Life Technologies, Paisley, UK) using a syringe, and cell suspension was centrifuged for 10 min at 200× *g*. The cell pellet was resuspended in RPMI-1640 supplemented with 10% of previously inactivated fetal bovine serum (FBS; Sigma-Aldrich^®^) and an antibiotic–antimycotic solution (penicillin 100 U/mL, streptomycin 100 µg/mL, and amphotericin B 0.25 µg/mL) (complete medium) with the addition of 30% L929 conditioned medium as a source of M-CSF. L929 cells were originally obtained from the American Type Culture Collection (ATCC, Rockville, MD, USA). The cells were distributed in 6-well plates and incubated at 37 °C and 5% CO_2_ for 6 days, changing the medium on day 4. The plates were washed with PBS twice to remove non-adherent cells, and then adherent cells were detached using a cell scraper and ice-cold RPMI-1640. The cells were centrifuged and resuspended in complete medium. The cells were counted in a Neubauer chamber, adjusted to 10^6^ cells/mL, and distributed in 96-well plates (100 µL/well) to be used in the experiments. An aliquot was used to confirm the generation of CD11b^+^F4/80^+^ bone marrow-derived macrophages (BMDM), using anti-mouse CD11b conjugated with PE (BD Biosciences, San Diego, CA, USA) and anti-mouse F4/80 conjugated with Bv401 (BioLegend, San Diego, CA, USA).

### 2.4. Peritoneal Cells Collection

After euthanasia, 5 mL of ice-cold PBS was injected in the mouse peritoneal cavity to wash the cavity. Peritoneal cells were recovered by centrifuging the collected fluid for 10 min at 200× *g*. The cells were resuspended in complete medium and counted in a Neubauer chamber. Cell density was adjusted to 10^6^ cells/mL, and the cells were distributed in 96-well plates (100 µL/well). The plates were incubated at 37 °C and 5% CO_2_. After 2 h, the plates were washed twice with PBS to remove non-adherent cells, and the remaining cells (peritoneal macrophages, PM) were incubated in complete medium. An aliquot of cells was used to confirm the macrophage phenotype, as described above.

### 2.5. Priming and Challenge of Macrophages with Heterologous Pathogens

After seeding BMDM and PM (1 × 10^6^ cells/mL; 100 µL/well), BCG or rBCG-S1PT was used to stimulate the cells at an MOI 0.1:1 (10^4^ CFU:10^5^ cells/well) for 24 h. The 24 h supernatant was collected to determine the “primary cytokine response”. Bacteria were removed by washing the cells with RPMI-1640, and new medium was added. The cells were left to rest in culture for 6 days, changing the medium on day 4. On day 7, the medium was refreshed (100 µL/well), and the cells were challenged with *C. albicans* (10^6^/mL), *S. aureus* (10^6^/mL), or LPS (10 ng/mL) (from *E. coli* O55:B5; Sigma-Aldrich^®^) for 24 h; then, the supernatants were collected to determine the “secondary cytokine response”.

### 2.6. Analysis of Cell Viability

Viability across the experiments was assessed using the FVS Viability Stain (BD Biosciences) according to the manufacturer’s instructions. Data were acquired using a FACS Canto II equipment and analyzed using FlowJo Software version 10 (BD Biosciences). The gating strategy is depicted in Appendix A.

### 2.7. Metabolism Analysis

Before the secondary challenge (after 6 days of resting), the supernatants were collected, and glucose and lactate were measured using a YSI Biochemical Analyzer (YSI Life Sciences, Yellow Springs, OH, USA).

### 2.8. Cytokine Production

The supernatant of cells challenged with *C. albicans*, *S. aureus*, or LPS (after 24 h of stimulation) or with culture medium alone (unchallenged control) was collected and frozen. The frozen samples were thawed, centrifuged, and the supernatants were assayed for inflammation-related cytokines (IL-6, IL-10, MCP-1, IFN-γ, TNF-α, and IL-12p70) using a Cytometric Beads Array (CBA mouse inflammatory kit, BD Biosciences), according to the manufacturer’s instructions. The assay lower limits of detection were between 2.5 and 17.5 pg/mL, depending on the cytokine, and the higher limit was 5000 pg/mL. Data were acquired using a FACS Canto II equipment and analyzed using FCAP Array Software (BD Biosciences). Production of IL-1β was measured by ELISA using the DuoSet^®^ Mouse IL-1β/IL-1F2 (R&D Systems, Inc., Minneapolis, MN, USA). The supernatant of cells before the challenge (after 6 days of resting) was also assayed for these cytokines to determine the activation status of resting cells.

### 2.9. In Vivo Generation of Innate Memory in Immunocompetent Mice

C57BL/6 mice were immunized subcutaneously with 10^6^ CFU of BCG or rBCG-S1PT and 30 days later were challenged with LPS (10 ng intravenously). Whole blood was collected 4 h later, and cytokine concentration in the serum was measured by flow cytometry using a CBA mouse inflammatory kit as previously described.

### 2.10. In Vivo Challenge with C. albicans of Immunodeficient SCID Mice

*C. albicans* (10^5^) was administered through the retroorbital vein to a single C57BL/6 mouse. After 7 days, both kidneys were recovered and plated onto YPD agar plates. A single colony was cultured in liquid YPD medium until reaching OD 0.6. The cells were counted in a Neubauer chamber, and 10^5^ *C. albicans* cells were used to inoculate another mouse. This procedure was repeated 4 times in order to increase virulence.

Immunodeficient NOD/SCID mice were administered intravenously 10^6^ CFU BCG or rBCG-S1PT. After two weeks, the mice were challenged with 10^5^ CFU of virulent *C. albicans*, and survival was followed for 8 weeks. Other groups of mice were euthanized 2 weeks after the *C. albicans* challenge, and their kidneys were collected and plated onto YPD agar in serial dilutions to evaluate CFU recovery.

### 2.11. Statistical Analysis

Comparison between groups in both in vitro and in vivo experiments was performed using a two-tailed Mann–Whitney U test. Difference in the survival curves was assessed by log-rank Mantel-Cox analysis; *p* values < 0.05 were considered statistically significant. Statistical analysis was performed using GraphPad Prism software version 7.04.

## 3. Results

### 3.1. In Vitro Model for Examining the Innate Memory Induced by rBCG-S1PT

In this study, we investigated whether rBCG-S1PT would confer innate memory on murine macrophages. The in vitro model to evaluate the generation of innate memory is depicted in Figure 1.

### 3.2. Primary Response of BMDM and PM to rBCG-S1PT

We first established the rBCG dose that would confer a proper stimulation without decreasing macrophage viability. Based on our results (Appendix A), we selected a MOI of 0.1:1 BCG per cell (e.g., 10^4^ CFU to 10^5^ cells), comparable to that in previous experiments using peripheral blood mononuclear cells (PBMC) [8].

BMDM and PM were exposed in vitro to medium alone, wild type BCG, or rBCG-S1PT. Macrophage activation (primary response) was determined 24 h after priming through the evaluation of IL-6, IL-10, MCP-1, IFN-γ, TNF-α, IL-12p70, and IL-1β. As expected, exposure of BMDM to BCG promoted the production of TNF-α and IL-6. When exposed to rBCG-S1PT, even higher levels of TNF-α, IL-6 IL-10, MCP-1, and IL-1β were produced (Figure 2A). Interestingly, priming peritoneal macrophages with BCG or rBCG-S1PT induced the production of TNF-α, IL-10, and IL-1β, whereas the spontaneous high levels of IL-6 were not further increased by either BCG strain (Figure 2B). The cytokines IL-12p70 and IFN-γ were detectable at very low levels and are not shown.

### 3.3. Priming with rBCG-S1PT Promotes a Higher Inflammatory Response to Challenge

Priming of BMDM and PM with BCG or rBCG-S1PT induced an increased production of inflammatory cytokines in response to the secondary heterologous challenge with *C. albicans*, *S. aureus*, or LPS (Figure 3). When assessing TNF-α, one of the main innate/inflammatory cytokines (upper panels), priming with BCG (striped bars) increased its production by both macrophage types only in response to *C. albicans* challenge. Conversely, rBCG-S1PT priming (black bars) induced a higher TNF-α production upon challenge with all agents (except in PM challenged with *S. aureus*) in comparison to both unprimed cells (white bars) and cells primed with BCG. When examining the production of another inflammatory cytokine, IL-6 (middle panels), cells primed with rBCG-S1PT afforded an increase in the response upon all heterologous challenges in comparison to medium and wild-type BCG (except BMDM challenged with *S. aureus*). BCG-primed peritoneal macrophages also exhibited an increased secondary IL-6 response in comparison to unprimed cells (when challenged with *S. aureus* and LPS), but at lower levels. The production of the anti-inflammatory cytokine IL-10 showed a dichotomy between BMDM and PM, with rBCG-S1PT priming essentially unable to induce IL-10 in BMDM, but increasing its production in PM challenged with *S. aureus* and *C. albicans*. The concomitant upregulation of inflammatory and anti-inflammatory mechanisms observed here suggests that the induction of higher inflammatory responses induced by rBCG-S1PT may be accompanied by regulatory mechanisms that prevent excessive/damaging inflammatory responses.

It should be noted that priming with rBCG-S1PT induced a macrophage activation that was not fully extinguished after one week (in terms of TNF-α and IL-6 production in the last 2 days of resting) (Appendix A). Challenging these cells resulted in an additional increase in cytokine production (Figure 3). Notably, in our experimental conditions, the two different macrophage types exhibited distinct patterns of general cytokine response. While BMDM appeared to be more sensitive to LPS, peritoneal macrophages reacted better to *S. aureus* and *C. albicans* (see white columns between mock and challenges in Figure 3 and note the different values on the vertical axis between BMDM and PM). The cytokines IL-12p70 and IFN-γ were detectable at very low levels and are not shown. Data for MCP-1 and IL-1β are shown in Appendix A. While no significant memory effect was detectable for IL-1β production, it is notable that the production of MCP-1 in BMDM (but not in PM) in response to LPS was significantly lower in BCG- and rBCG-primed cells, in a typical tolerance type of memory.

### 3.4. rBCG-S1PT Induces Metabolic Changes in Memory Macrophages

Previous studies reported a shift towards the glycolytic pathway in memory macrophages. Here, priming with rBCG-S1PT promoted a higher consumption of glucose, linked with the production of lactate in macrophages left to rest for 6 days after priming. In BMDM, a significant shift in the lactate/glucose ratio was observed (Figure 4A). In peritoneal macrophages, rBCG-S1PT did not induce a greater consumption of glucose in memory cells, yet a higher lactate concentration was observed, with a consequent significant shift of the lactate/glucose ratio (Figure 4B).

### 3.5. rBCG-S1PT Induces an Increased Cytokine Response upon LPS Challenge In Vivo

To test whether rBCG-S1PT could enhance the response to a heterologous challenge in vivo, C57BL/6 mice were subcutaneously administered saline (control), BCG, or rBCG-S1PT and challenged intravenously with LPS 4 weeks later. The cytokine levels in the serum were measured before and 4 h after the challenge. Mice primed with BCG displayed an increased production of TNF-α upon LPS stimulation, while the production of other cytokines was comparable to that of controls (Figure 5). In rBCG-S1PT-primed mice, a priming-dependent increase was evident for TNF-α, IL-1β, and IFN-γ. Notably, rBCG-S1PT-primed mice reacted to LPS with an increase in the circulating levels of IFN-γ that was significantly higher than that observed in BCG-primed mice.

### 3.6. Previous Immunization with rBCG-S1PT Increases Protection against C. albicans Challenge

BCG has been shown to induce non-specific protection against heterologous pathogens. Here, we evaluated whether rBCG-S1PT could afford this non-specific protection in vivo. SCID mice were administered saline, BCG, or rBCG-S1PT and challenged 2 weeks later with *C. albicans*. Survival was followed for 8 weeks after the challenge. Both BCG and rBCG-S1PT improved mice survival in comparison to the saline group (Figure 6A). When mice were evaluated for CFU in the kidneys one week after the challenge, a significant reduction in CFU was observed in the group primed with rBCG-S1PT (mean 51.5 × 10^3^ CFU/kidney) in comparison to that treated with saline (mean 260.2 × 10^3^ CFU/kidney), whereas the effect of BCG was not significant (mean 203.6 × 10^3^ CFU/kidney) (Figure 6B).

## 4. Discussion

In this study, we demonstrate that a recombinant BCG vaccine, rBCG-S1PT, is able to induce an innate immune memory response. To date, few agents have been shown to induce innate immune memory, and each of them seems to induce a different profile. BCG, LPS, and β-glucan are often used in experimental conditions [12,18,19,20], but only BCG is used as a human vaccine and has shown beneficial nonspecific effects in clinical [21] and epidemiological settings [22]. BCG encompasses a variety of molecules that may trigger innate immunity through several pathways. For instance, many components of the mycobacterial cell wall such as peptidoglycan, arabinogalactan, and mycolic acids are sensed by TLR2 and TLR4 [23]. Specific proteins such as Ag85 can directly act as a TLR agonist [24]. Additionally, as a live attenuated vaccine, BCG viability is a factor that may also affect its protective efficacy [25].

The intrinsic adjuvanticity of BCG, i.e., its capacity to non-specifically amplify the immune response, is one of the reasons supporting the generation of hundreds of recombinant strains in the last 30 years [9]. rBCG-S1PT was initially devised as a neonatal vaccine against pertussis [2]. It protects adult and neonate mice against *B. pertussis* challenge, exhibiting an increased Th1-predominant immune response in this animal model. When repurposed to immunotherapy in a mouse bladder cancer orthotopic model, rBCG-S1PT showed a greater reduction of the bladder’s tumor weight and an increased survival time [5]. The anti-tumor effect was related to a higher expression of TNF-α and IL-10 in the bladder [6]. In the present study, BMDM stimulated with rBCG-S1PT showed a marked increase in the production of IL-6, TNF-α, IL-10 MCP-1, and IL-1β in comparison to cells stimulated with BCG. Conversely, the primary response induced by rBCG-S1PT in peritoneal macrophages showed increased IL-1β but lower TNF-α and IL-10 production in comparison to that induced by wild-type BCG. In previous studies, our group showed that human PBMC stimulated with rBCG-S1PT produce increased levels of IL-8, IL-6, and IL-10, but similar levels of IL-1β and TNF-α in comparison to cells stimulated with wild-type BCG [8]. This highlights that immune activation induced by rBCG-S1PT is different in distinct cell populations.

In our experimental conditions, BMDM and peritoneal macrophages exposed to rBCG-S1PT maintained a significant cytokine production even after one week from the initial stimulation, differently from the extinction of the primary response in macrophages exposed to wild-type BCG. This suggests that rBCG-S1PT can promote a higher and more prolonged inflammatory response.

Re-stimulation of rBCG-S1PT-primed macrophages with *S. aureus*, *C. albicans*, and LPS further increased their response, in terms of TNF-α and IL-6 production. In BMDM, both wild-type BCG and rBCG-S1PT appeared to induce the classical innate memory “recall” response (primary response, extinction of activation, memory response), and the recombinant strain generated a sustained unique response. Conversely, the type of response observed in peritoneal macrophages may be considered a non-classical memory response, in which primed cells acquire a long-lasting activation status that can be further upregulated upon subsequent challenges. This type of innate memory is well known in invertebrates with the name of “sustained unique response” or “acquired resistance” [26,27,28,29,30]. Thus, rBCG-S1PT, and likely other rBCG strains, may differ from wild-type BCG in the type of innate memory and cell reprogramming they generate. Additionally, the cytokine response of peritoneal macrophages was overall higher than in BMDM. This is an important observation, as it shows that different macrophage populations (as for instance macrophages residing in different organs) can develop different innate memory profiles to the same combination of priming/challenge agents. This observation underlines the importance of a local, organ-targeted induction of innate memory in future preventive or therapeutic interventions.

While innate memory generation was evident in macrophages after heterologous challenge, the Warburg effect (enhanced aerobic glycolysis with lactate production, generally associated with innate memory generated by β-glucan) was not clearly observed. This may depend on the fact that memory induced by BCG is not necessarily associated with the Warburg effect, since BCG can concomitantly increase both glycolysis and oxidative phosphorylation [31]. Notably, however, the Warburg effect was evident in rBCG-S1PT-primed BMDM, while it was absent in BCG-primed cells. This observation confirms the dissociation between Warburg effect and innate memory induced by BCG (no Warburg effect in BCG-primed cells).

We sought to investigate if administration of rBCG-S1PT would induce innate immune memory responses in vivo. Mice immunized with BCG or rBCG-S1PT produced higher and similar amounts of TNF-α in the serum upon LPS challenge. Interestingly, only rBCG-S1PT was able to induce significant levels of IL-1β and IFN-γ, which are inflammatory cytokines important for the establishment of innate memory in humans and mice, respectively [32,33]. The increased production of IFN-γ confirms previous data reporting a Th1-biased effect of rBCG-S1PT in comparison to wild-type BCG [2]. It is interesting to note that IFN-γ was only induced in rBCG-S1PT-primed mice upon re-stimulation with LPS, while negligible levels were detected before the challenge.

BCG immunization was shown to protect mice against *C. albicans* infection, assessed both as CFU burden and as survival [12]. Our results confirm the non-specific protection induced by BCG against *C. albicans*. To better understand the role of rBCG-S1PT in inducing innate immune memory, we administered BCG or rBCG-S1PT to SCID mice, which lack T and B cell responses, and challenged them with *C. albicans* 2 weeks later. Immunization with rBCG-S1PT caused a mean reduction of 80% in the *C. albicans* burden in the kidneys compared to untreated mice, while wild-type BCG only caused a non-significant reduction of 20%. Previous studies reported a 10-fold reduction in *C. albicans* CFU in the kidneys of SCID mice immunized with BCG [12]. These differences may be due the distinct BCG strain (Danish versus Moreau) or the dose used (750 µg/mice versus 10^6^ CFU/mice). Overall, however, no significant difference in survival was observed between mice treated with rBCG-S1PT and those treated with the parental BCG strain, of which both reduced mortality to a similar extent. The observation of BCG-induced non-specific protection in experimental models has been variable, also in the case of epidemiological data [22]. For instance, BCG-induced innate immune memory can increase efferocytosis of alveolar phagocytes against H1N1 [34], but this does not translate into protection of mice against H7N9 challenge [15]. Furthermore, as already mentioned, the innate memory effects are likely different among the organs, and their generation in specific locations may strongly depend on the route of administration and the dosage of the memory-inducing agent [15,35]. How innate memory is generated is an issue that is being actively discussed and investigated [10,36].

Recombinant BCG strains can maintain the immunostimulatory characteristics of wild-type BCG on the innate immune response, while able to induce adaptive immunity against BCG and recombinant antigens. It is therefore expected that rBCG vaccines could stimulate both protective innate and adaptive immune responses [17]. To date, no rBCG vaccine has been experimentally evaluated for the induction of non-specific immune memory. The closest vaccine evaluated was the *M. tuberculosis*-based attenuated vaccine (MTBVAC), an improved TB vaccine candidate. MTBVAC is in phase II clinical trial and demonstrated protection of mice [37] and rhesus macaques against *M. tuberculosis* aerosol challenge [38]. Previous immunization with MTBVAC can also enhance the immune response in synergy with the pertussis vaccine [39], while having the same capacity as BCG to induce innate memory [40].

The BCG vaccine has been used for a century, and even today new features are still being discovered. The suggestion that innate immune memory could explain why BCG can offer protection against unrelated diseases brought a renewed attention to this vaccine. Moreover, BCG is being repurposed to treat other diseases, and studies investigating its effect against SARS-CoV-2 are under evaluation [41,42,43]. The expression of heterologous antigens in BCG should maintain the original characteristic and can introduce new ones, as demonstrated in this work with rBCG-S1PT. Here, we provide experimental evidence that rBCG-S1PT is able to increase innate immune memory and, ultimately, non-specific protection.

## 5. Conclusions

In this work, we demonstrated that rBCG-S1PT can enhance the innate immune memory response and induce an increased non-specific protection against *C. albicans* challenge. Further studies to phenotype the involved macrophage population and/or address the role of other innate cells can provide additional evidence on the mechanisms involved. To our knowledge, this is the first report describing the capacity of a recombinant BCG strain to afford improved innate immune memory responses.

## Figures and Tables

**Figure 1 vaccines-10-00234-f001:**
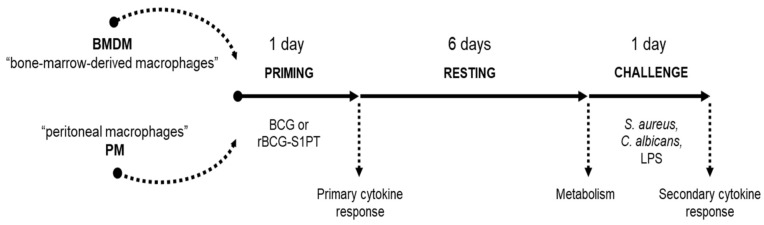
Schematic representation of the experimental approach using BMDM and PM. C57BL/6 mice were euthanized, and femur, tibia, and peritoneal cells were collected. Bone marrow cells were differentiated to macrophages (BMDM). Peritoneal macrophages (PM) were purified by adherence. BMDM and PM were primed with BCG or rBCG-S1PT for 1 day, and the supernatants were collected to determine the primary cytokine response. The cells were then left to rest for 6 days. Before the addition of the secondary stimuli (challenge), the supernatant was collected to measure glucose and lactate levels (metabolism analysis). After 1 day of re-stimulation with *S. aureus*, *C. albicans*, or LPS, the supernatants were collected to determine the secondary cytokine response.

**Figure 2 vaccines-10-00234-f002:**
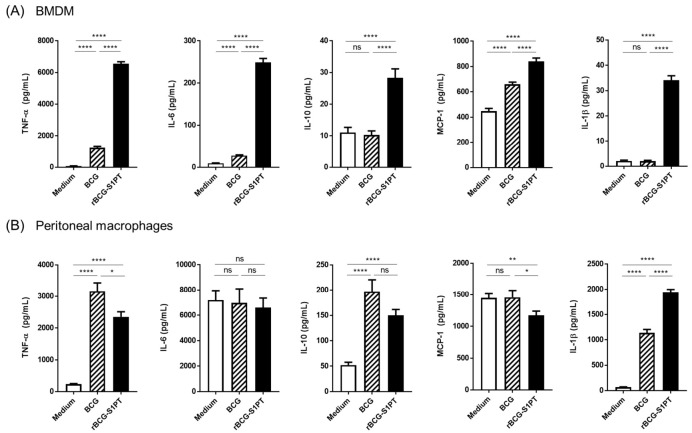
Primary response of BMDM (**A**) and peritoneal macrophages (**B**) to wild-type BCG or rBCG-S1PT. Bone marrow-derived macrophages and peritoneal macrophages recovered from naïve mice were exposed in vitro to culture medium alone (medium), BCG, or rBCG-S1PT (MOI 0.1:1) for 24 h. The production of TNF-α, IL-6, IL-10, MCP-1, and IL-1β (panels from the left to right) was measured in the 24 h supernatants. Statistical significance was assessed via Mann–Whitney U test. * *p* < 0.05, ** *p* <0.01, **** *p* <0.0001, ns = not significant. Bars represent mean ± SEM of 10 replicate samples for the medium and BCG groups and of 12 replicate samples for the rBCG-S1PT group.

**Figure 3 vaccines-10-00234-f003:**
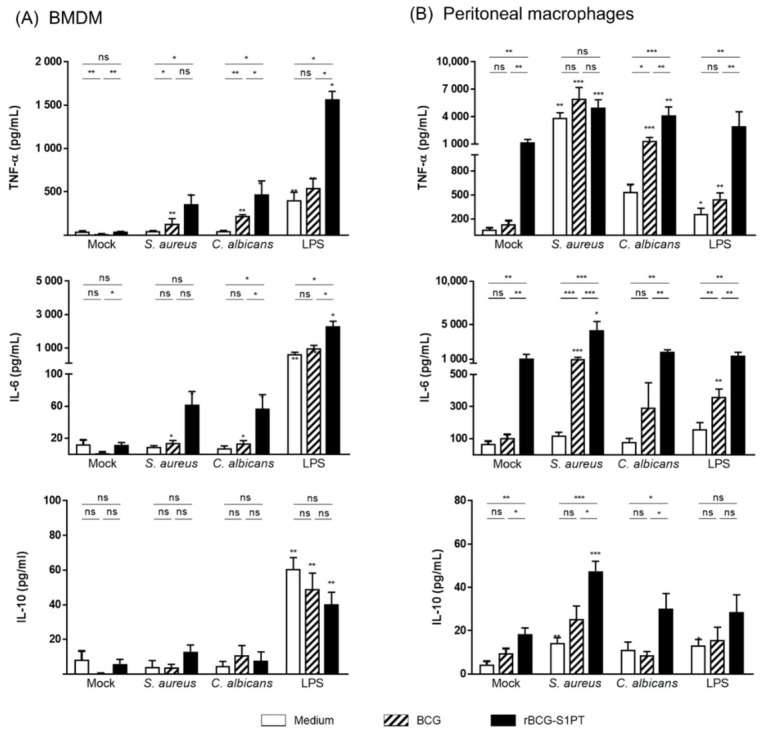
Memory response of primed macrophages to heterologous challenges. BMDM (**A**) and peritoneal macrophages (**B**) from naïve mice were exposed in vitro to culture medium alone (unprimed, white bars), BCG (striped bars), or rBCG-S1PT (black bars) (MOI 0.1:1) for 24 h. Cells were then left to rest for 6 days and re-stimulated with medium alone (mock) or with *S. aureus*, *C. albicans*, or LPS (horizontal axis). The production of TNF-α (upper panels), IL-6 (middle panels), and IL-10 (lower panels) was measured after 24 h. Statistical analysis was performed via Mann–Whitney U test. * *p* < 0.05, ** *p* < 0.01, *** *p* < 0.001, ns = not significant. Bars represent mean ± SEM of 4–5 (for BMDM) and 5–8 (for peritoneal macrophages) replicate samples. Asterisks over the columns in *S. aureus*, *C. albicans*, and LPS refer to the comparison with the respective mock.

**Figure 4 vaccines-10-00234-f004:**
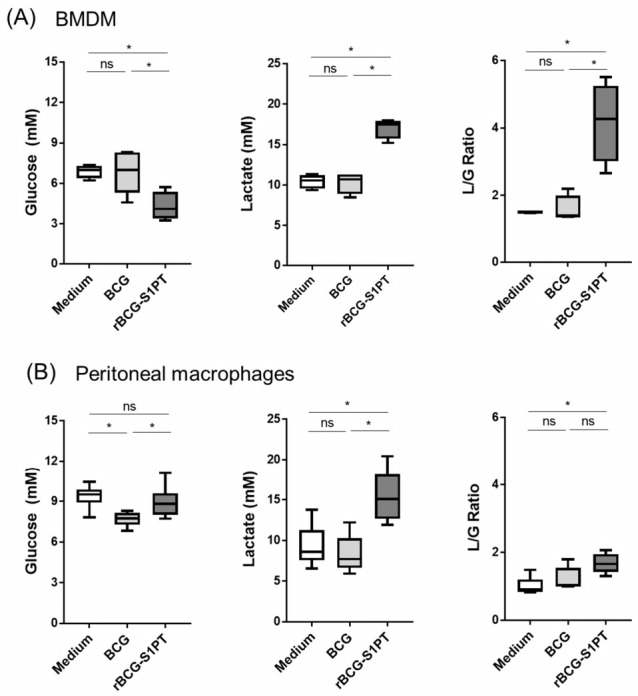
rBCG-S1PT induces metabolic changes in memory macrophages. BMDM (**A**) and PM (**B**) from naïve mice were exposed in vitro to culture medium alone (medium), BCG, or rBCG-S1PT (MOI 0.1:1), then left to rest in fresh medium for 6 days. Glucose and lactate levels and the lactate/glucose (L/G) ratio were compared between groups. Statistical analysis was performed via Mann–Whitney U test. * *p* < 0.05, ns = not significant. Boxes and whiskers represent value distribution with median (horizontal line), first, and third quartile (box limits) and minimum and maximum values (whiskers) of four replicate samples for BMDM and six replicate samples for PM.

**Figure 5 vaccines-10-00234-f005:**
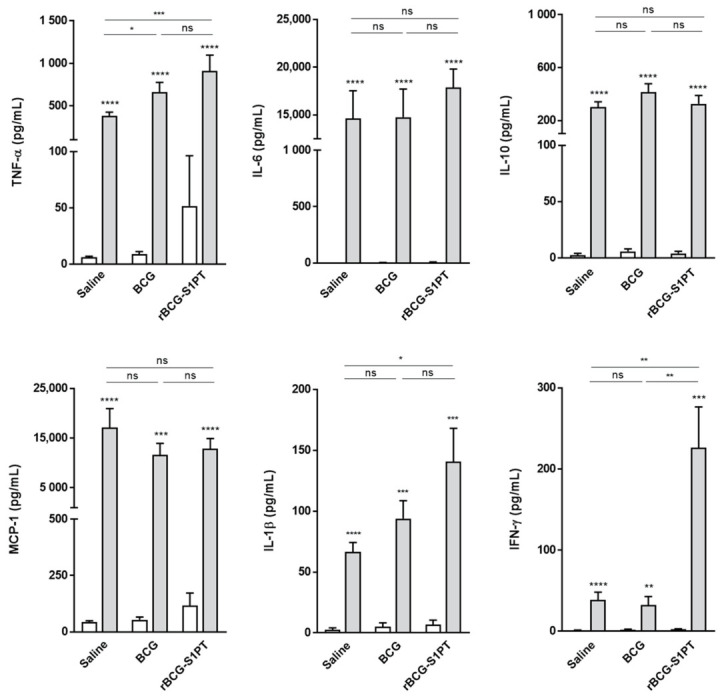
Priming of mice with rBCG-S1PT enhances the cytokine response to LPS challenge. Mice were administered saline or 10^6^ CFU of BCG or rBCG-S1PT and challenged with LPS 4 weeks later. The cytokine levels in the serum were determined before (white bars) and 4 h after the challenge (gray bars). Statistical analysis was performed via Mann–Whitney U test. * *p* <0.05, ** *p* <0.01, *** *p* < 0.001, **** *p* < 0.0001, ns = not significant. Bars represent mean ± SEM of values from 10 mice. Asterisks over gray columns represent the comparison with the respective control group (white column).

**Figure 6 vaccines-10-00234-f006:**
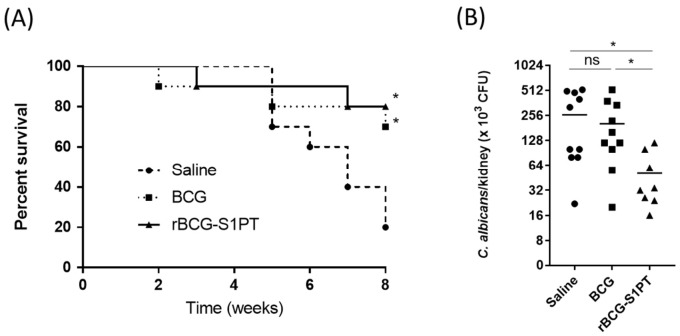
Priming with rBCG-S1PT increases the protection against *C. albicans* challenge. (**A**) Groups of SCID mice (n = 10/group) were primed with 10^6^ CFU of BCG or rBCG-S1PT and challenged 2 weeks later with *C. albicans*. The mice were followed for survival for 8 weeks after infection. (**B**) Using other groups of mice (n = 5/group), kidneys were collected one week after the challenge, and *C. albicans* CFU was evaluated. The results are representative of two and three experiments, respectively. The statistical significance of survival was determined via Log-rank (Mantel-Cox) test, and CFU via Mann–Whitney U test. * *p* < 0.05 vs. saline control. ns = not significant. Bars represent the mean ± SEM of values from the kidneys of five mice.

## Data Availability

The data presented in this study are available on request from the corresponding author.

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
