# Peer review of "Recombinant BCG Expressing the Subunit 1 of Pertussis Toxin Induces Innate Immune Memory and Confers Protection against Non-Related Pathogens"

_vaccines, 2022, doi:10.3390/vaccines10020234_

Round 1

Reviewer 1 Report

In their manuscript, Kanno and collaborators evaluated the induction of innate immune memory by a recombinant BCG strain expressing the genetically detoxified S1 subunit of the pertussis toxin (rBCG-S1PT). The authors found that mouse peritoneal macrophages exposed to the modified BCG increased their innate response to a challenge from unrelated pathogens and observed a potential innate-memory dependent control of C. albicans infection. The authors conclude that recombinant BCG could potentially increase innate memory and non-specific protection more effectively than the wild type BCG.

The manuscript is extremely clear, well presented and very well written and does not require any modification other than minor typos here and there.

Author Response

We thank the reviewer for the positive appraisal.

Reviewer 2 Report

Kanno et al describe their experiments, causing and modulating innate immune responses in bone marrow and peritoneal macrophages based on BCG or rBCG-S1PT vaccination studies. These ex vivo / in vitro studies are complemented by in vivo studies in normal and NOD/SCID mice, showing the benefit of the vaccination in non-BCG infection situations in a memory setting. The experiments are comprehensive and straight-forward, the manuscript is well readable. I have only minor concerns.

Minor issues:

Lines 67,68: Please check / correct citation.

Line 93: "…added with 30% L929 conditioned medium as a source of M-CSF." Please add reference to the used medium or to the source of the L929 cells.

Line 162: "…unpaired nonparametric two-tailed Mann-Whitney t test." Please check / correct the sentence according to the used test. It should be either a t test or a Mann-Whitney U test. Please provide the reference of the software used to compute the statistical analysis.

Lines 199, 200, 226, 255, 272, 293: Please check / correct the sentence according to the used test. It should be either a t test or a Mann-Whitney U test.

Line 254: "Production of glucose and lactate and the lactate/glucose (L/G) ratio were compared between groups." Whether glucose is produced or not cannot be drawn from the provided results. Please edit or check whether the following might be applicable: "Glucose and lactate levels and the lactate/glucose (L/G) ratio were compared between groups."

Figure 2, 3, 4: Please provide the exact number n for each group.

The in vivo studies well complement the in vitro functional analyses. It might be hypothesized -yet however is not proven and obviously beyond the scope of this study- that the in vivo effects are mediated through macrophages as well. The authors should briefly discuss potential experiments in the future to better phenotype the mentioned macrophages, e.g. by mRNA expressional profiling and possibilities to trace those cells in vivo for longitudinal studies. This could at least hypothetically bridge the gap between the functional studies ex vivo and the observations in vivo.

Author Response

We appreciate the reviewers’ comments and suggestions, which helped improve the manuscript. We present bellow a point-by-point reply.

1- Lines 67,68: Please check / correct citation.

The citation format was corrected.

2- Line 93: "…added with 30% L929 conditioned medium as a source of M-CSF." Please add reference to the used medium or to the source of the L929 cells.

We thank the reviewer for calling our attention to this. The L929 cells were kindly provided by Dr. Momtchilo Russo from the Universidade de São Paulo, which in turn obtained them from the American Type Culture Collection (ATCC). The sentence in the manuscript was modified to “…added with 30% L929, originally obtained from the American Type Culture Collection (ATCC, Rockville, Maryland, USA) conditioned medium as a source of M-CSF.” (lines 96-97). We also add Dr. Momtchilo Russo and Eliane Gomes in the acknowledgments section for the kind donation of the L929 cell line (lines 414-415).

3- Line 162: "…unpaired nonparametric two-tailed Mann-Whitney t test." Please check / correct the sentence according to the used test. It should be either a t test or a Mann-Whitney U test. Please provide the reference of the software used to compute the statistical analysis.

The sentence was reformulated to “…using a two-tailed Mann-Whitney U test.” (line 165) and a new sentence was added “Statistical analysis was performed using GraphPad Prism software version 7.04.” (lines 166-167).

4- Lines 199, 200, 226, 255, 272, 293: Please check / correct the sentence according to the used test. It should be either a t test or a Mann-Whitney U test.

Reference for the Mann-Whitney U test was corrected in all figure legends (lines 202, 230, 259, 277, 297).

5- Line 254: "Production of glucose and lactate and the lactate/glucose (L/G) ratio were compared between groups." Whether glucose is produced or not cannot be drawn from the provided results. Please edit or check whether the following might be applicable: "Glucose and lactate levels and the lactate/glucose (L/G) ratio were compared between groups."

We agree and the sentence was corrected as suggested (lines 257-258).

6- Figure 2, 3, 4: Please provide the exact number n for each group.

In Figure 2, the sentence was amended and now reads “Bars represent mean ± SEM of 10 replicate samples for medium and BCG groups and 12 replicate samples for the rBCG-S1PT group.” (line 203).

In Figure 3, there were 5 samples in BMDM-stimulated groups. For PM-stimulated groups, we used 6-8 samples. Differences in the number of samples are due to limitations in the number of cells acquired from mice (both BMDM and PM). These cells required a primary and a secondary stimulus which further decreases the number of replicate samples. This particular experiment (depicted in figure 3) was performed 3 times. (line 231)

Figure 4, the sentence was modified to “4 replicate samples for BMDM and 6 for PM.” (line 261).

7- The in vivo studies well complement the in vitro functional analyses. It might be hypothesized -yet however is not proven and obviously beyond the scope of this study- that the in vivo effects are mediated through macrophages as well. The authors should briefly discuss potential experiments in the future to better phenotype the mentioned macrophages, e.g. by mRNA expressional profiling and possibilities to trace those cells in vivo for longitudinal studies. This could at least hypothetically bridge the gap between the functional studies ex vivo and the observations in vivo.

We thank the reviewer for the suggestions. A new phrase was added: “Further studies to phenotype these macrophages and/or address the role of other innate cells can provide additional evidence on the mechanisms involved in the increased innate memory response induced by rBCG-S1PT.” (lines 407-409).

Reviewer 3 Report

In this study, the authors evaluated the induction of innate memory by a recombinant BCG strain expressing the genetically detoxified S1 subunit of the pertussis toxin (rBCG-S1PT). They demonstrated that the recombinant BCG vaccine, rBCG-S1PT, is able to induce an innate immune memory response. So, the paper can be accepted for the publication after some minor revisions.

  1.  In the introduction section, add more motivation and novelties of your study. 
  2. Give the compassion between your results and the others existing in the literature.
  3. There are some typos. The authors should carefully read the manuscript.
  4. Add a conclusion section of your paper.
  5. Check and unify the citation of the references

Author Response

We appreciate the reviewers’ comments and suggestions, which helped improve the manuscript. We present bellow a point-by-point reply.

1-In the introduction section, add more motivation and novelties of your study. 

The paragraph describing previous immunomodulation properties of rBCG-S1PT was modified (in bold).

Recombinant BCG expressing the genetically detoxified S1 subunit of the pertussis toxin (rBCG-S1PT) was demonstrated to induce protection in mice against Bordetella pertussis challenge. Its pathogen-specific response was consistently demonstrated in adult and neonate mice (2-4). The distinct immunostimulating capacity of rBCG-S1PT was evident in different experimental models. For instance, rBCG-S1PT showed an enhanced immune activation and greater anti-tumor effect in a mouse model of bladder cancer – a known clinical application of wild type BCG (5, 6). In an experimental model of asthma, previous immunization with rBCG-S1PT can enhance the Th1 lung immunity and modulate the allergic response induced by OVA (7). Additionally, rBCG-S1PT can increase the inflammatory response in human peripheral blood mononuclear cells in vitro (8). (lines 39-45).

2- Give the compassion between your results and the others existing in the literature.

We have included a phrase to compare our results on BCG effect to previous literature (line 368) Furthermore, we had a discussion on possible comparison with MTBVAC (lines 389-394). However, it is important to note that even BMDM and PM in the same model showed different primary and secondary responses. In this context, the use of different cell populations prevents a direct comparison with MTBVAC, which used human monocytes. The closest experiment was performed previously in our group using human peripheral mononuclear cells which showed an increased response of IL-6, IL-8 and IL-10 when incubated with rBCG-S1PT.

3- There are some typos. The authors should carefully read the manuscript.

The manuscript was revised thoroughly, and minor typos and grammatical errors were corrected.

4- Add a conclusion section of your paper.

We agree that a conclusion section is interesting, although the format proposed by the journal does not include it. Therefore, we have rearranged the text and included a conclusion section (lines 404-411).

5- Check and unify the citation of the references

A missing reference in methods section (Nascimento et al., 2009) was not according to MDPI’s instruction and was corrected.